# From Speculation Detection to Trustworthy Relational Tuples in Information Extraction

**Kuicai Dong**[1,2]**, Aixin Sun**[1]**, Jung-Jae Kim**[2]**, Xiaoli Li**[1,2,3]

[1] School of Computer Science and Engineering, Nanyang Technological University, Singapore
kuicai001@e.ntu.edu.sg, axsun@ntu.edu.sg
[2] Institute for Infocomm Research, A*STAR, Singapore
[3] A*STAR Centre for Frontier AI Research, Singapore
{jjkim, xlli}@i2r.a-star.edu.sg

## Abstract

Speculation detection is an important NLP task to understand text factuality. However, the extracted speculative information (*e.g.,* speculative polarity, cue, and scope) lacks structure and poses challenges for direct utilization in downstream tasks. Open Information Extraction (OIE), on the other hand, extracts structured tuples as facts, without examining the certainty of these tuples. Bridging this gap between speculation detection and information extraction becomes imperative to generate *structured speculative information* and *trustworthy relational tuples*. Existing studies on speculation detection are defined at sentence level; but even if a sentence is determined to be speculative, not all factual tuples extracted from it are speculative. In this paper, we propose to study speculations in OIE tuples and determine *whether a tuple is speculative*. We formally define the research problem of *tuple-level speculation detection*. We then conduct detailed analysis on the LSOIE dataset which provides labels for speculative tuples. Lastly, we propose a baseline model *SpecTup* for this new research task.[1]

## 1 Introduction

The concept of speculation is closely connected to modality, which has been extensively explored in both linguistics and philosophy (Saurí, 2008). Modality is defined as the expression of the speaker's level of commitment to the events mentioned in a text (Saurí et al., 2006). Other related terms, such as "*hedging*", "*evidentiality*", "*uncertainty*", and "*factuality*", are also used when discussing different aspects of speculation. It is argued that information falling under speculation cannot be presented as *factual* information (Morante and Daelemans, 2009). In NLP applications, *factual information* plays a critical role in comprehending

---

[1]We will release the source code upon paper acceptance.

sentences or documents. Consequently, the identification of speculative information, as potentially opposed to facts, has garnered considerable interest from NLP community (Szarvas et al., 2008; Konstantinova et al., 2012; Ghosal et al., 2022).

The conventional task of speculation detection involves two subtasks: cue identification and scope detection, which are typically performed at the sentence level. Speculation cue refers to a minimal linguistic unit that indicates speculation (*e.g.,* "*might*", "*plans to*", "*subject to*"). Speculation scope is the text fragment governed by the corresponding cue in a sentence. Consider the following example sentence: "*The UN **plans to** [release a report].*", the speculative cue "***plans to***" affects the meaning of the scope "*release a report*". Note that the speculation, along with its cue and scope, is often unstructured. **Question I** arises: Can unstructured speculative information be directly applied to downstream tasks?

To extract structured information, we can leverage on Open Information Extraction (OIE). OIE aims to generate relational factual tuples from unstructured open-domain text (Yates et al., 2007). The extracted tuples are in form of ($ARG_0$, *Relation*, $ARG_1$, ..., $ARG_n$), also called facts. By definition, OIE system is domain-independent and highly scalable, thus allowing users to obtain facts with low cost. The extracted factual tuples are beneficial to many downstream tasks such as question answering (Khot et al., 2017), knowledge base population (Martínez-Rodríguez et al., 2018; Gashteovski et al., 2020), and text summarization (Fan et al., 2019). Here, **Question II** arises: *Shall we trust all relational tuples extracted by OIE as facts?* Apparently, the answer is no, as not all sentences state facts, partly due to speculation. Current studies on OIE do not consider speculation, which can lead to the use of unreliable tuples as facts, affecting the accuracy of downstream applications.

To mitigate issues related to **Question I** and **II**,

we propose to serialize speculative information into relational tuples. In such a way, we can empower information extraction with speculation detection to produce *trustworthy relational tuples* by identifying *structured speculative information*. However, there are typically multiple relational tuples in one sentence. A sentence that contains speculative information does not necessarily mean that all its tuples are speculative. Also, the scopes defined in existing speculation datasets do not align well with the tuples extracted by OIE. Therefore, current sentence-level speculation detection cannot be applied to OIE. To bridge this gap, we propose a new research task that focuses on *tuple-level* speculation detection. In simple words, we aim to indicate whether a tuple is speculative. In other words, the speculation scope becomes a tuple of OIE.

To the best of our knowledge, there is no dataset specifically designed for tuple-level speculation detection. Nevertheless, the recently released LSOIE (Solawetz and Larson, 2021) provides a timely preview of this interesting research task. LSOIE is a large-scale OIE dataset converted from the QA-SRL 2.0 dataset (FitzGerald et al., 2018). We observe that LSOIE provides additional annotation to some OIE tuples with *modal auxiliary verbs*. Table 1 lists a few example sentences and their annotated OIE tuples with speculation. The modal auxiliary verbs include '*might*', '*can*', '*will*', '*would*', '*should*', and '*had*'; these words express the meaning of possibility, ability, intention, past intention, suggestion, and past event, respectively. It is important to note that *these modal auxiliary verbs annotated on* OIE relations **may (43.3%)** or **may not (56.7%)** appear in the original sentences of the LSOIE dataset. In this study, we use these modal auxiliary verbs as speculation labels for relational tuples, as they express the degree of uncertainty at tuple level.

Tuple-level speculation detection is challenging, because it is common for only a certain portion of a sentence to carry speculative semantics. Certain words (*e.g.,* "may", "if", "plan to"), also known as speculation cues, are responsible for semantic uncertainty, making part of a sentence (or the corresponding extracted tuples) vague, ambiguous, or misleading. In this paper, we develop a simple yet effective baseline model to detect **Spec**ulation at **Tup**le level (called **SpecTup**). SpecTup detects speculation from two perspectives: *semantic* and *syntactic*. To model relation-aware semantics,

SpecTup adds additional relation embedding into BERT transformers, and uses BERT's hidden state of the tuple relation word as *semantic representation*. For syntactic modeling, SpecTup explicitly models the sub-graph of dependency structure of input sentence, which includes immediate neighbours of the tuple relation word. It adaptively aggregates nodes in the sub-graph using a novel relation-based GCN, and uses the aggregated representation as *syntactic representation*. The concatenated semantic and syntactic representations are then used for speculation detection.

Our contributions in this paper are threefold. **First**, we propose a new research task to detect *tuple-level* speculation. This task links speculation detection and information extraction. It examines the reliability of relational tuples, which aligns well with the goal of OIE to extract only *factual* information. **Second**, we conduct a detailed analysis on the tuple-level speculation labels from two aspects: (i) their presence in language, and (ii) the level of detection difficulty. **Third**, we propose SpecTup, a baseline model to detect tuple-level speculation. SpecTup leverages both semantic (BERT) and syntactic (Sub-Dependency-Graph) representations. We perform extensive experiments to analyze the research task of tuple-level speculation, and our results show that SpecTup is effective.

## 2 Tuple-level Speculation Analysis

We first review the annotation processes of the QA-SRL Bank 2.0 and LSOIE datasets, with a key focus on tuple-level speculation labels. We then study the distribution of speculation labels, by the perceived level of difficulty in detection.

### 2.1 Annotation

QA-SRL Bank 2.0 dataset consists of question-answer pairs for modeling verbal predicate-argument structure in sentence (FitzGerald et al., 2018).[2] A number of questions are crowdsourced for each verbal predicate in a sentence, and each answer corresponds to a contiguous token span in the sentence. Examples of QA-SRL annotations can be found in Appendix A.2. Crowdworkers are required to define questions following a 7-slot template, *i.e.,* Wh, Aux, Subj, Verb, Obj, Prep, Misc. Among them, 'Aux' refers to auxiliary verbs,[3] and

---

[2]Question-Answer Driven Semantic Role Labeling. https://dada.cs.washington.edu/qasrl/

[3]There are three main auxiliary verbs: '*be*', '*do*', '*have*'. Besides them, there's a special type that affects grammatical

| ID | Example sentence | OIE Tuple with **speculation** | Meaning |
|---|---|---|---|
| 1 | Adults were allowed to opt out of using computers. | (*adults*, **can** *opt*, *using computers*) | ability |
| 2 | It is unclear if the suspects left with any property. | (*suspects*, **might** *left*, *any property*) | possibility |
| 3 | The UN plans to release a final report in two weeks. | (*the UN*, **will** *release*, *a final report*) | intention |
| 4 | Gargling with warm salt water are reasonable. | (*warm salt water*, **should** *gargling*) | suggestion |

Table 1: Examples of speculation annotation in LSOIE dataset. The speculation labels are in boldface as part of tuple relation. To facilitate understanding, we underline the speculation cues in the example sentence, and elaborate the meaning of speculation label under the 'Meaning' column. Note that a sentence usually contains multiple tuples, we truncate the long sentence and demonstrate only the tuple with speculation for conciseness.

| Subset | #Sent | #Tuple | #Spec. Tuple | %Spec |
|---|---|---|---|---|
| wiki$_{test}$ | 4,670 | 10,635 | 1,015 | 9.5% |
| wiki$_{train}$ | 19,630 | 45,931 | 4,110 | 8.9% |
| sci$_{test}$ | 6,669 | 11,403 | 1,569 | 13.8% |
| sci$_{train}$ | 19,193 | 33,197 | 4,337 | 13.1% |
| Total | 50,162 | 101,166 | 11,031 | 10.9% |

Table 2: Number of sentences, tuples, tuples with speculation, and the percent of tuples with speculation.

| Example sentence | Category |
|---|---|
| The UN will release a report. | Easy |
| The UN will recently release a report. | Med |
| The UN plans to release a report. | Hard |

Table 3: Examples of 3 cases of speculation according to detection difficulty. All three sentences convey the same fact: (*the UN*, **will** *release*, *a report*).

answers to questions with modal auxiliary verbs reflect speculative information of the fact as well. Note that these modal verbs may or may not explicitly appear in the original sentence. Hence, they are provided based on the annotator's understanding of the sentences.

The QA-SRL Bank 2.0 dataset is then converted to a large-scale OIE dataset (LSOIE) by Solawetz and Larson (2021). LSOIE defines $n$-ary tuples, in the form of ($ARG_0$, *Relation*, $ARG_1$, ..., $ARG_n$) in two domains, *i.e.,* Wikipedia and Science. During the conversion, the modal auxiliary verbs in the QA-SRL questions are retained in tuple relations, as shown in Table 1. In this study, we consider these modal auxiliary verbs to reflect speculation. Consequently, in the LSOIE dataset, **tuples with speculation** are those whose relation contains any of the following six modal auxiliary verbs: '*might*', '*can*', '*will*', '*would*', '*should*', and '*had*'. In this work, we follow Lobeck and Denham (2013) to interpret 6 types of speculation labels as follows: '*can*' shows or infers general ability; '*will*' and '*would*' are used to show intention or to indicate certainty; '*might*' shows possibility; '*should*' is used to suggest or provide advice; '*had*' refers to past actions or events.

Table 2 reports the statistics of sentences, tuples, and the tuples with speculation in the LSOIE

dataset.[4] Overall, 10.9% of the ground truth tuples contain speculation, indicating the substantial presence of speculative 'facts', some of which are non-factual. However, as no OIE system considers the speculation information, a considerable number of unreliable facts are being extracted, without any speculative tags on them. We thus propose to develop an model that tags speculation on the extracted tuples.

To the best of our understanding, neither the crowdsourcing process of QA-SRL Bank 2.0 nor the conversion of LOSIE specifically focuses on speculation, as it is not the primary focus of these two datasets. Without explicit focus, existing annotations truly reflect the crowdworker's natural understanding, and there are no bias towards any specific type of speculation. We manually examine a large number of the speculative labels and observe that labels are indeed reliable enough to be suitable for evaluating tuple-level speculations.

## 2.2 Perceived Level of Detection Difficulty

By analyzing the speculative labels, we have grouped them into three categories, as listed in Table 3, based on the perceived level of detection difficulty. (i) The **easy** category refers to cases where the speculation label (*e.g.,* "*will*") literally

---

mood (*e.g.,* '*will*', '*might*'), called *modal auxiliary verbs*.

[4]We notice that some sentences in the Wiki subset appear again (or repeated) in the Sci subset. In this study, we remove the repeated sentences from the Sci subset. Therefore, our reported numbers differ from those in the original paper.

| Subset | #Spec. Tuple | Easy | Med | Hard |
|---|---|---|---|---|
| wiki$_{test}$ | 1,015 | 20.0% | 23.6% | 56.4% |
| wiki$_{train}$ | 4,110 | 20.0% | 22.3% | 57.6% |
| sci$_{test}$ | 1,569 | 22.2% | 19.1% | 58.8% |
| sci$_{train}$ | 4,337 | 22.7% | 22.4% | 54.9% |
| Total | 11,031 | 21.3% | 22.0% | 56.7% |

Table 4: Distribution of the tuples with speculation by difficulty level.

appears in the sentence and is located immediately beside the tuple relation (*e.g.,* "*will release*"). (ii) In **medium** category, the speculation label is present in the sentence, but it is not located directly beside the tuple relation (*e.g.,* "*will recently release*"). (iii) The **hard** category refers to the cases where the speculation label is not present in the sentence at all (*e.g.,* "*plans to*" means "*will*").

Table 4 illustrates the distribution of speculation labels across the three levels of difficulty. As seen, 56.7% of speculative tuples in the LSOIE dataset fall under the hard category. It is evident that detecting these hard cases of speculation is challenging, requiring deep semantic understanding of both the input sentence and the corresponding fact. Additionally, we present a more fine-grained breakdown of the difficulty distribution of speculation labels for each label in Appendix A.3.

## 3 Task Formulation

As discussed, there are no existing OIE systems consider speculation when extracting tuples. To fully leverage the capabilities of existing OIE models, it is more meaningful and practical to formulate speculation detection as a *post-processing* task, where the goal is to determine whether a tuple extracted from a sentence is speculative.

Formally, the inputs are the source sentence that is split into words $w_i$ (or tokens) $s = [w_1, \ldots, w_n]$, and a set of relational tuples $T = \{t_1, \ldots, t_m\}$ extracted from this sentence.[5] Sentence-level speculation detection makes one-pass prediction on sentence $s$. In comparison, the task of *tuple-level speculation detection* is to predict whether a tuple $t_i$ is speculative (*i.e.,* a binary classification task), based on $t_i$ and its source sentence $s$. We focus on tuple-level speculation detection in this paper.

---

[5] Each tuple $t_i$ is represented by its components $t_i = [x_1, \ldots, x_l]$ where one $x$ is the relation and the rest $x$'s are arguments. Each $x$ corresponds to a contiguous span of words $[w_j, \ldots, w_{j+k}]$.

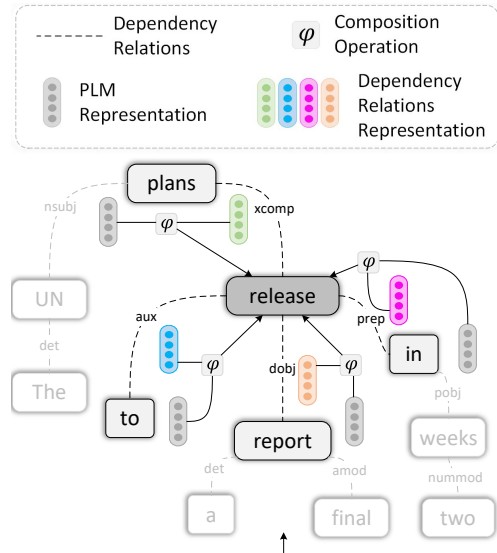

Figure 1: The relation-based aggregation of SpecTup. The tuple relation 'release' adaptively aggregates its neighbours based on different dependency relations.

As the tuples set $T$ typically contains multiple tuples, sentence $s$ will be analyzed multiple times with different $t_i$. We assume that the relation (or relation word) of a tuple is different from the relations of all the other tuples from the same sentence and thus that a tuple can be represented with its relation. Note that the task can be extended to predict if a tuple is speculative and, if speculative, which of the six modal auxiliary verbs indicates the speculation type of the tuple.

## 4 Method: SpecTup

The proposed baseline consists of three modules: **pre-trained BERT** to encode semantic representation of input sequence (in Section 4.1), **GCN encoder** to capture the syntactic representation (in Section 4.2), and **linear layer** for binary/multi-class speculation detection (in Section 4.3).

Given an example tuple (*the UN*, *release*, *a report*) in Figure 1, the words "*plans to*" in its source sentence, which modifies the relation word "*release*", indicate the speculation of the tuple. Note that such a cue of the speculation of a tuple is typically syntactically related to the relation word of the tuple in its source sentence. We thus encode the dependency structure of a source sentence, as exemplified in Figure 1, as well as the word sequence of the sentence for the speculation detection task.

### 4.1 Semantic (Word Sequence) Representation

We leverage BERT (Devlin et al., 2019) as token encoder to obtain token representations. Specifically, we project all words $[w_1, \ldots, w_n]$ into embedding space by summing their word embeddings[6] and tuple relation embeddings:

$$h_i = \boldsymbol{W_{word}}(w_i) + \boldsymbol{W_{rel}}(tuple\_rel(w_i)) \quad (1)$$

where $W_{word}$ is trainable and initialized by BERT word embeddings. $W_{rel}$ is a trainable tuple relation embedding matrix, and the function $tuple\_rel(w_i)$ returns 1 if the word $w_i$ is the tuple relation of the instance; otherwise, 0. Specifically, $W_{rel}$ initializes all words into binary embeddings (positive embedding for relation word, and negative embedding to non-relation words). It is worth noting that, we only utilize the tuple's relation word for speculation detection and not its arguments. Therefore, the model explicitly emphasizes the difference between relation tokens and non-relation tokens.

Then, we use $h_s = [h_1, \ldots, h_n]$ as the input to BERT encoder and utilize BERT's last hidden states as semantic representations:

$$h_i^{sem} = \text{BERT}(h_i) \in \mathbb{R}^{d_h} \quad (2)$$

### 4.2 Syntactic (Dependency Graph) Representation

**Relation Sub-graph Modelling.** Speculation cue refers to a minimal unit that indicates speculation *e.g.,* "*might*", "*plans to*", "*subject to*". However, such cue words are not explicitly labelled in the source sentence. We assume that a speculative tuple is the outcome of *a speculation cue in the sentence directly modifying the tuple's relation word*. Therefore, we model the words that are syntactically related to the tuple's relation word $v$, through the sentence's dependency structure.

Specifically, we extract a sub-dependency-graph $N(v)$ from the sentence's dependency structure, which consists of the immediate (or one-hop) neighbours of the tuple relation word $v$. In $N(v)$, each node $u$ directly links to $v$ with an associated dependency relation $r$, as highlighted by the clear (or non-blurry) lines in Figure 1, where $(u, r)$ denotes the link (or edge), omitting $v$ since all links share the common word node $v$. We call this sub-dependency-graph as 'sub-graph' and the whole

---

dependency graph of the input sentence as 'full-graph', and compare their impact on the speculation detection task in Section 5.3.

[6]If a word maps to multiple sub-words by BERT tokenization, we average the sub-word representations.

**Relation-aware GCN.** Inspired by CompGCN (Vashishth et al., 2020), we devise a strategy to embed each dependency relation as a vector and aggregate the corresponding neighbouring nodes together. The representation of word $v$, denoted by $h_v$, is updated:

$$h_v^{syn} = f\left( \sum_{(u,r) \in N(v)} \varphi(u, r) \boldsymbol{W_r} h_u^{sem} \right) \quad (3)$$

where $f(\cdot)$ is the activation function, $\boldsymbol{W_r} \in \mathbb{R}^{d_h \times d_h}$ is a trainable transformation matrix. $\varphi(u, r)$ is the neighbour connecting strength, computed based on the dependency type $r$:

$$\varphi(u, r) = h_u^{sem} \cdot \boldsymbol{W_{dep}}(r) \quad (4)$$

where $\cdot$ is the dot production operator. $\boldsymbol{W_{dep}} \in \mathbb{R}^{d_h \times N_{dep}}$ is a trainable matrix. $N_{dep}$ is the number of unique dependency relations.

### 4.3 Speculation Detection

Finally, we concatenate the semantic representation in Equation (2) and the syntactic representation from GCN in Equation (3) as follows:

$$h_v^{final} = h_v^{sem} \oplus h_v^{syn} \quad (5)$$

where $h_v^{final}$ is used by the classification layer to perform speculation detection.

For the binary classification task of identifying if a tuple is speculative in the source sentence or not, we use binary cross-entropy loss:

$$L_{CE} = -\frac{1}{N} \sum_{i=1}^{N} y_i \log(p_i) + (1 - y_i)\log(1 - p_i) \quad (6)$$

where $N$ is the number of training instances. $y_i$ is the gold standard label, and $p_i$ is the Softmax probability for the $i^{th}$ training instance.

For the multi-class classification task of classifying speculative tuples into the 7 classes (non-speculative and the six modal auxiliary verbs), we use the multi-class cross-entropy loss:

$$L_{CE_{\text{multi}}} = -\frac{1}{N} \sum_{i=1}^{N} \sum_{j=1}^{7} y_{i,j} \log(p_{i,j}) \quad (7)$$

## 5 Experiments

### 5.1 Experiment Setup

**LSOIE Dataset.** To avoid potential bias introduced by any specific OIE model, in our experiments, we only use tuples from the LSOIE dataset as inputs to the proposed tuple-level speculation detection model. For the example sentence *"The UN plans to release a report"*, the tuple would be (*the UN*, *release*, *a report*). Our task is to determine whether this tuple is speculative and, if speculative, which of the six modal auxiliary verbs indicates the speculation type. In our experiments, we combine two subsets (Wikipedia and Science) into one big dataset. As the result, we have 38,823 sentences and 79,128 tuples for training, originally from $\text{wiki}_{\text{train}}$ and $\text{sci}_{\text{train}}$ sets. We have 11,339 sentences and 22,038 tuples for testing, originally from the $\text{wiki}_{\text{test}}$ and $\text{sci}_{\text{test}}$ sets.

**Evaluation Metrics.** We use *Precision, Recall, and $F_1$ measures* for evaluation. We report the results of the binary classification task from the following three perspectives:

**(1) Macro-averaged scores.** This is the unweighted average over the two classes, *i.e.,* speculative and non-speculative tuples. Micro-average is not used because nearly 89.1% of tuples are non-speculative, which would dominate the measures.

**(2) Positive scores.** This is the set of measures computed for the positive (*i.e.,* speculative) class.

**(3) Recall by difficulty.** This is recall of the speculative tuples by the perceived difficulty level (Section 2.2): the percentage of easy, medium, and hard cases that are correctly detected. [7]

### 5.2 Baselines

As this is a new research task, there are no existing baselines for comparison. We have designed four sets of methods to compare: (i) only semantic representations, (ii) only syntactic representations, (iii) both semantic and syntactic representations, and (iv) keywords matching.

**Semantic-Only.** We leverage on BERT to get the semantic representations of the source sentence $s$. Note that all tokens in the sentence are encoded with tuple relation embedding (see Equations 1 and 2). As a result, the tuple relation information is explicitly fused to all source tokens. We then evaluate the effectiveness of using weighted pooling of all tokens in sentence ($\text{SEM}_{\text{sentence}}$), in tuple ($\text{SEM}_{\text{tuple}}$), and in tuple relation ($\text{SEM}_{\text{relation}}$) for speculation detection.

**Syntactic-Only.** Based on the sentence dependency parsing, we evaluate $\text{SYN}_{\text{full-graph}}$ and $\text{SYN}_{\text{sub-graph}}$. The former uses a GCN to aggregate node information from the entire dependency graph, while the latter only aggregates the subgraph with one-hop neighbors of the relation word, as described in Section 4.2.

**Semantic and Syntactic.** We combine both semantic and syntactic representations and implement two methods. The first one is $\text{SEM}_{\text{sentence}}$ + $\text{SYN}_{\text{full-graph}}$ that uses semantic embedding and dependencies of all tokens in the sentence. In comparison, SpecTup leverages only the embeddings and sub-graph dependencies of tuple relation token, which is equivalent to $\text{SEM}_{\text{relation}}$ + $\text{SYN}_{\text{sub-graph}}$.

**Keywords Matching in Dependency sub-graph.** Besides the neural baselines mentioned above, we also experiment with simple keywords matching methods. For these methods, we use a pre-defined speculative keywords dictionary. A tuple is classified as speculative if any of its immediate neighbours in dependency parse tree is one of the words in the dictionary. We first use the 6 modal verbs as the dictionary. Then we additionally include the most frequent speculative keywords (top 10, 20, and 30). Speculative keywords selection is described in details in Appendix A.4.

### 5.3 Main Results

Table 5 reports the experimental results of the proposed baseline methods.

**Semantic vs Syntactic.** The three baselines using semantic representations significantly outperform the two baselines using syntactic representations. This highlights that the semantic information is more crucial for the model to understand speculation. We also observe that the recall scores of syntactic models are comparable to those of semantic models. By combining both semantic and syntactic information, SpecTup outperforms all baselines, particularly in recall and $F_1$ scores.

**Semantic:** Tuple Relation vs Full Sentence / Tuple. For the baselines with semantic representations only, the baseline using tuple relation performs not

---

[7]Precision is not applicable here, as the task is not to predict the difficulty level of each tuple.

| Models | Macro-averaged | | | Positive | | | Recall by difficulty | | |
|---|---|---|---|---|---|---|---|---|---|
| | $Pr$ | $Re$ | $F_1$ | $Pr$ | $Re$ | $F_1$ | Easy | Medium | Hard |
| **Keyword Matching** (Dep sub-graph) | | | | | | | | | |
| Modal Verbs | 79.6 | 66.6 | 70.6 | 67.4 | 35.7 | 46.6 | 99.8 | 59.9 | 3.6 |
| Modal Verbs + Top 10 speculative words | 79.3 | 72.3 | 75.1 | 65.4 | 48.0 | 55.4 | 99.8 | 60.8 | 24.5 |
| Modal Verbs + Top 20 speculative words | 77.8 | 72.2 | 74.6 | 62.4 | 48.4 | 54.5 | 99.8 | 60.8 | 25.3 |
| Modal Verbs + Top 30 speculative words | 77.4 | 72.2 | 74.4 | 61.6 | 48.5 | 54.3 | 99.8 | 60.8 | 25.5 |
| **Semantic-only** (BERT) | | | | | | | | | |
| $SEM_{sentence}$ | **86.6** | 72.0 | 76.9 | **80.0** | 45.6 | 58.1 | 99.8 | 46.4 | 25.6 |
| $SEM_{tuple}$ | 84.0 | 73.2 | 77.1 | 76.0 | 47.9 | 58.8 | 99.8 | 57.6 | 23.2 |
| $SEM_{relation}$ | 84.5 | 73.1 | 77.3 | 75.7 | 48.3 | 59.0 | 99.8 | 64.4 | 23.4 |
| **Syntactic-only** (Dependency) | | | | | | | | | |
| $SYN_{sub\text{-}graph}$ | 72.3 | 70.2 | 71.0 | 53.2 | 46.6 | 49.7 | 95.4 | 41.0 | 35.5 |
| $SYN_{full\text{-}graph}$ | 72.8 | 70.0 | 71.2 | 54.3 | 46.2 | 49.9 | 95.6 | 42.5 | 34.7 |
| **Semantic** and **Syntactic** | | | | | | | | | |
| $SEM_{sentence}$ + $SYN_{full\text{-}graph}$ | 82.4 | 74.0 | 77.8 | 70.5 | 52.3 | 60.1 | 99.8 | 58.0 | 32.8 |
| $SEM_{relation}$ + $SYN_{full\text{-}graph}$ | 81.1 | 75.4 | 78.3 | 67.2 | 57.0 | 61.7 | **100** | 62.3 | 37.5 |
| $SEM_{relation}$ + $SYN_{sub\text{-}graph}$ (SpecTup) | 80.7 | **77.5** | **79.0** | 66.9 | **58.9** | **62.6** | **100** | **67.2** | **40.9** |

Table 5: Main results for binary tuple speculation detection. The best results are in boldface.

worse than the other two baselines using more information (tuple and full sentence), suggesting that some relations are more likely to be speculative than others, and the BERT encoding has implicitly considered the contextual information.

**Syntactic: Sub-graph vs Full-graph.** In terms of syntactic modeling, the baselines using subgraphs are slightly better than those using fullgraphs, indicating that it is valuable to encode speculation cues with the other immediate neighbours of the tuple relation.

**Neural Methods vs Keywords Matching** Observe that SpecTup outperforms all keywords matching methods by a large margin. In particular, the recall of hard case speculations in SpecTup is nearly double that of keyword matching, indicating the advantages of semantic and syntactic modelling through neural networks.

Overall, SpecTup achieves the best results. However, the $F_1$ of speculative class is only 62.6, leaving a big room for future investigation.

## 6 Multi-class Speculation Classification

As mentioned in Section 3, we extend the task to predict both the existence of speculation and its specific type, as defined by the 6 auxiliary modal verbs: '*might*', '*can*', '*will*', '*would*', '*should*', and '*had*'. In this way, we perform multi-class classification among 7 classes: non-speculative and 6 types of speculative classes.

| Spec Type | #N | $Pr$ | $Re$ | $F_1$ | % Hard |
|---|---|---|---|---|---|
| Non-Spec | 19,456 | 93.5 | 97.7 | 95.6 | - |
| Spec | 2,616 | 66.7 | 46.6 | 54.9 | 57% |
| – can | 1,003 | 87.6 | 47.9 | 61.9 | 40% |
| – might | 720 | 53.9 | 39.0 | 45.3 | 94% |
| – will | 339 | 82.4 | 59.3 | 69.0 | 38% |
| – should | 283 | 39.4 | 56.2 | 46.3 | 82% |
| – would | 170 | 85.3 | 37.6 | 52.2 | 28% |
| – had | 101 | 94.4 | 33.7 | 49.6 | 36% |

Table 6: Breakdown results of multi-class speculation classification by speculation class.

### 6.1 Results Overview

As reported in Table 6, the multi-class $F_1$ score of speculative classes decreases to 54.9%, compared to the positive $F_1$ score of 62.6% in Table 5. As expected, detecting speculative type is much more challenging than the existence of speculation.

Table 6 also reports the break-down precision, recall, $F_1$ scores of the 6 different speculative classes. The performance of these classes are determined mainly by two factors: (i) The hard cases are naturally more challenging, because the speculation labels do not appear in source sentences. They can only be inferred based on the semantics. We observe that '*might*' class consists of 94% of hard cases, leading to the lowest $F_1$ score 45.3% among all speculative classes. (ii) The performance is affected by the number of instances. The '*had*' class is a minor class, with a very low $F_1$ score 49.6%, even when only 36% of its labels are hard cases.

| ID | Example Sentence | Spec Cue | OIE Tuple with Speculation |
|---|---|---|---|
| 1 | The UN plans to release a final report. | plans to | (*the UN*, **will** *release*, *a final report*) |
| 2 | The UN plans to reduce troops. | plans to | (*the UN*, **might** *reduce*, *troops*) |
| 3 | Charlemagne planned to continue the tradition. | planned to | (*Charlemagne*, **would** *continue*, *the tradition*) |

Table 7: Three examples of speculative tuples in LSOIE with similar speculation cue ("*plan to*"), but with different speculation labels. We truncate the long sentence and demonstrate only the tuple with speculation for conciseness.

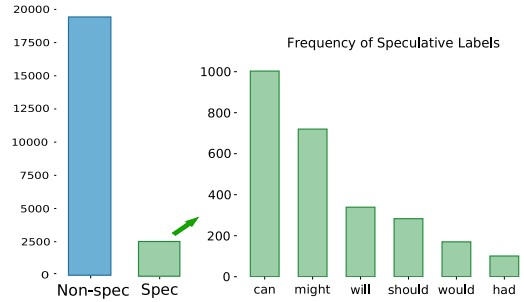

Figure 2: Distribution Non-spec and spec labels, and the break-down distribution of different speculative types.

## 6.2 Long-tail Distribution

The labels exhibit a long-tail class distribution. As shown in Figure 2, 10.9% tuples are speculative. Among them, '*might*' and '*can*' contribute to 65.9% of the speculative labels. Other speculative types are associated with relatively small number of samples. Such class imbalance is a key source of performance degradation in classification. For comparison, the $F_1$ scores of head class '*Non-spec*' and tail class '*Spec*' are 95.6% and 54.9%, respectively (with a big difference of 40.7%). Among speculative classes, the $F_1$ scores of head class '*can*' and tail class '*had*' are 61.9% and 49.6%, respectively (with a big difference of 12.3%), although the percent of hard cases of '*can*' is even higher than that of '*had*' (40% > 36%).

## 6.3 Ambiguous Speculative Types

The annotation of speculative types can be ambiguous, due to subtle difference between speculation labels. Table 7 lists three example sentences, all containing "*plan to*", which can express both likelihood and intention. Different annotators might interpret the sentence in different ways, thus annotating the same speculation cue with different labels. Manual checking of annotations in LSOIE shows that such ambiguous labels are rather common, particularly among hard cases. Such ambiguous hard cases are more challenging to distinguish.

## 7 Related Work

**Speculation Detection.** Speculation detection from text is essential in many applications in information retrieval (IR) and information extraction (IE). Research in computational linguistics has made significant advances in detecting speculations. Remarkably influential datasets include the BioScope Corpus for uncertainty and negation in biomedical publications (Szarvas et al., 2008), the CoNLL 2010 Shared Task (Farkas et al., 2010) for detecting hedges and their scope in natural language texts, unifying categorizations of semantic uncertainty for cross-genre and cross domain uncertainty detection (Szarvas et al., 2012), and the SFU Review Corpus (Konstantinova et al., 2012) for negation, speculation and their scope in movie, book, and consumer product reviews.

Current work on speculation is to detect the existence of speculation and/or the speculation cue/scope in a given *sentence*. The speculation scope is the maximum number of words affected by the phenomenon. The speculation cue refers to the minimal unit which expresses speculation. For instance, the most frequent 4 keywords ("*if*", "*or*", "*can*", and "*would*") contribute to 57% of the total speculation cues in SFU Review Corpus. We argue that the detection of such speculation cue at sentence-level is relatively easier. In comparison, *tuple-level* speculation cue detection requires not only locating the correct speculative keywords, but also assigning them to the correct tuples. It is critical for many real-world applications relying on updated and accurate tuples in knowledge base.

**Traditional and Neural OIE systems.** Open Information Extraction (OIE) was first proposed by Yates et al. (2007). Before deep learning era, many statistical and rule-based systems have been proposed, including Reverb (Fader et al., 2011), Clausie (Corro and Gemulla, 2013), and Stanford OpenIE (Angeli et al., 2015), to name a few. These models extract relational tuples based on hand-crafted rules or statistical methods. The extraction

mainly relies on syntactic analysis.

Recently, neural OIE systems have been developed and showed promising results. Neural OIE systems can be roughly divided into two types: generative and tagging-based (Zhou et al., 2022). Generative OIE systems (Cui et al., 2018; Kolluru et al., 2020a; Dong et al., 2021) model tuple extraction as a sequence-to-sequence generation task with copying mechanism. Tagging-based OIE systems (Stanovsky et al., 2018; Kolluru et al., 2020b; Dong et al., 2022) tag each token as a sequence tagging task. Most neural OIE models are designed for end-to-end training and the training data are mostly silver data generated by traditional systems.

## 8    Conclusion

Speculation detection, which is essential in many applications in IR and IE, has not been explored in OIE. So we formally define a new research task to perform *tuple-level* speculation detection on OIE extracted tuples. We notice that the LSOIE dataset, although not dedicated for speculation detection, provides us a timely preview of this interesting research task. We conduct a detailed analysis on the speculative tuples of the LSOIE dataset. To provide a glimpse of this research task, we develop a simple but effective model named SpecTup to detect tuple-level speculation. As an emerging research area, we believe there is a big room for further exploration, from dataset construction to more effective models.

## Acknowledgments

This research is supported by the Agency for Science, Technology and Research (A*STAR) under its AME Programmatic Funding Scheme (Project #A18A2b0046 and #A19E2b0098).

## Limitations

We analyze the limitations of our work from three aspects as follows.

**Annotation Quality of Speculation.**    Annotating speculation is challenging due to the subtle difference between different speculative types. As discussed in Section 2.1, neither the crowdsourcing of QA-SRL Bank 2.0 nor the conversion of LOSIE pays specific attention to speculation. The annotation of auxiliary modal verbs in the question is based on crowd workers' natural understanding of the sentences. Without explicit focus on speculation, the existing annotations reflect the crowd worker's natural understanding without any bias towards specific focusing on speculation. Therefore, we argue that the annotation is suitable for evaluating tuple-level speculations.

**Quality of OIE Tuples.**    We use the ground truth tuples of LSOIE as inputs, to avoid potential bias introduced by any specific OIE model (see Section 5.1). However, existing OIE systems are not so perfect, as we note that the state-of-the-art $F_1$ score in LSOIE dataset is 0.71 (Vasilkovsky et al., 2022). The tuple-level speculation detection, as a post-possessing task, will inevitably suffer from the imperfectly extracted tuples. However, SpecTup can largely mitigate such issue. SpecTup only relies on the tuple relation for the speculation detection, rather than taking the full tuple with all arguments. Tuple relation is usually a verb or a verbal phrase that are straightforward to obtain. Therefore, so long as an OIE system can extract correct tuple relations, SpecTup can make predictions accordingly.

**Modelling Speculation Cues.**    We model the immediate neighbours of the tuple relation in dependency parsing as speculation cues (see Section 4.2). However, some speculation cues are not the immediate neighbours of the tuple relation. However, considering the full dependency tree leads to poorer results. We leave it for future work to explore a better way to effectively model speculation cues.

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

# A Appendix

## A.1 Implementation Detail

We build and run our system with Pytorch 1.9.0 and AllenNLP 0.9.0 framework. The dependency annotations are by spacy-transformers. [8] There are in total 45 types of dependency labels. We tokenize and encode input sentence tokens using bert-base-uncased. [9] The experiments are conducted with Tesla V100 32GB GPU and Intel® Xeon® Gold 6148 2.40 GHz CPU. The experimental results are averaged over 3 runs with different random seeds. Each epoch is around 15 minutes on a single Tesla V100 32GB GPU. The hidden dimension $d_h$ for semantic representation $h_i^{sem}$ and that of syntactic representation $h_i^{syn}$ are both 768.

## A.2 QA-SRL Annotation

QA-SRL stands for Question-Answer (QA) driven Semantic Role Labeling (SRL). It is a task formulation which uses question-answer pairs to label verbal predicate-argument structure (FitzGerald et al., 2018). Given a sentence $s$ and a verbal predicate $v$ from the sentence, annotators are asked to produce a set of *wh*-questions that contain $v$ and whose answers are phrases in $s$. The questions are constrained in a template with seven fields, $s \in$ **Wh**×**Aux**×**Subj**×**Verb**×**Obj**×**Prep**×**Misc**, each associated with a list of possible options. Answers are constrained to be a subset of words in the sentence but not necessarily have to be contiguous spans. Table 8 shows example annotations of an input sentence "*The UN plans to release a final report in two weeks.*". In its first annotated question: "*When will someone release something*" contains a verb and its answer is a phrase "*in two weeks*" in the sentence. The answer tells us that "*in two weeks*" is an argument of "*release*". Enumerating all such pairs provides a relatively complete representation of the verb's arguments and modifiers.

## A.3 Fine-grained Breakdown of Difficulty Distribution

Table 9 presents a more fine-grained breakdown of the difficulty distribution of speculation labels for each label. It shows that 92.5% of '*might*' labels and 81.5% of '*should*' labels belong to hard category, while the distribution of other labels is

---

[8] https://spacy.io/universe/project/spacy-transformers
[9] https://huggingface.co/bert-base-uncased

| Verbal predicate | 7-slot Question | | | | | | | Answer |
|---|---|---|---|---|---|---|---|---|
| | Wh | Aux | Subj | Verb | Obj | Prep | Misc | |
| release | When | will | someone | release | something | - | - | in two weeks |
| | What | will | someone | release | - | - | - | a final report |
| | Who | will | - | release | something | - | - | The UN |
| plans | Who | - | - | plans | - | to do | something | The UN |
| | What | does | someone | plan | - | to do | - | release a final report |

Table 8: The QA-SRL annotations for a newswire sentence: *"The UN **plans** to **release** a final report in two weeks."*

more even. This can help use understand better the challenges of speculation detection task.

### A.4 Speculative Keywords

Speculation keyword or cue is the minimal unit which expresses speculation (Szarvas et al., 2008).

Konstantinova et al. (2012) annotate the Simon Fraser University Review corpus [10] with negation, speculation, and their cues. This corpus consists of 400 user reviews from *Epinions.com*. The 10 most frequent speculative keywords in the SFU Review Corpus are listed in Table 10.

Szarvas et al. (2012) select three corpora (*i.e.,* BioScope, WikiWeasel, and FactBank) from different domains, *e.g.,* biomedical, encyclopedia, and newswire. The BioScope corpus (Szarvas et al., 2008) contains clinical texts as well as biological texts from full papers and scientific abstracts; the texts are manually annotated for hedge cues and their scopes. The WikiWeasel corpus (Farkas et al., 2010) is annotated for weasel cues and semantic uncertainty, from randomly selected paragraphs taken from Wikipedia pages. The FactBank is a newswire dataset (Saurí and Pustejovsky, 2009). Events are annotated in the dataset and they are evaluated on the basis of their factuality from the viewpoint of their sources. The 20 most frequent speculative keywords in the BioScope, WikiWeasel, and FactBank are also shown in Table 10.

**Speculative Keywords Matching.** As discussed in Section 5.2, we consider keyword matching as simple baselines. The matching relies on a pre-defined speculative keywords dictionary. A tuple is classified as speculative if its immediate neighbours in dependency parsing tree contain any word in the dictionary. We first use the 6 auxiliary modal verbs ('*might*', '*can*', '*will*', '*would*', '*should*', and '*had*') as the dictionary. Shown in Table 5, modal verbs based matching leads to very low recall, as

expected. We then include more speculative keywords in our keywords dictionary.

Specifically, we sum the frequency of all frequent cues across four datasets in Table 10, and keep the 30 most frequent cues as exemplified in Table 11. We then build three variants on top of the basic dictionary, by adding the top 10, 20, and 30 speculative cues to the dictionary. The enriched dictionary largely increases the recall scores (see Table 5). We notice that including top 10 keywords significantly increases the recall of speculative class by 12.3%, and recall of hard case speculations by 20.9%. In comparison, including additional top 11-20 keywords only marginally increases the recall of speculative class by 0.4%, and recall of hard case speculations by 0.8%. Furthermore, the increase of including additional top 21-30 keywords is negligible.

---

[10] http://www.sfu.ca/~mtaboada/SFU_Review_Corpus.html

| Type / Diff. | can | | might | | will | | should | | would | | had | |
|---|---|---|---|---|---|---|---|---|---|---|---|---|
| | Num | (%) | Num | (%) | Num | (%) | Num | (%) | Num | (%) | Num | (%) |
| Easy | 1,111 | 36.1 | 114 | 3.9 | 512 | 36.3 | 84 | 8.1 | 349 | 39.7 | 189 | 34.4 |
| Medium | 1,404 | 33.0 | 104 | 3.6 | 384 | 27.2 | 108 | 10.4 | 267 | 30.3 | 162 | 29.5 |
| Hard | 1,743 | 40.9 | 2,676 | 92.5 | 516 | 36.5 | 845 | 81.5 | 264 | 30.0 | 198 | 36.1 |
| Total | 4,258 | 100 | 2,894 | 100 | 1,412 | 100 | 1,037 | 100 | 880 | 100 | 549 | 100 |

Table 9: Number and percentage (%) of speculation labels by class and by difficulty level.

| SFU | | BioScope | | FactBank | | WikiWeasel | |
|---|---|---|---|---|---|---|---|
| Cue | Frequency | Cue | Frequency | Cue | Frequency | Cue | Frequency |
| if | 876 | suggest | 810 | expect | 75 | may | 721 |
| or | 820 | may | 744 | if | 65 | if | 254 |
| can | 765 | indicate | 404 | would | 50 | consider | 250 |
| would | 594 | investigate | 221 | may | 43 | believe | 173 |
| could | 299 | appear | 213 | could | 29 | would | 136 |
| should | 213 | or | 197 | possible | 26 | probable | 112 |
| think | 211 | possible | 185 | whether | 26 | suggest | 108 |
| may | 157 | examine | 183 | believe | 25 | possible | 93 |
| seem | 150 | whether | 169 | likely | 24 | allege | 81 |
| probably | 121 | might | 155 | think | 24 | likely | 80 |
| - | - | can | 117 | might | 23 | might | 78 |
| - | - | likely | 117 | will | 21 | seem | 67 |
| - | - | could | 112 | until | 16 | think | 61 |
| - | - | study | 101 | appear | 15 | regard | 58 |
| - | - | if | 99 | seem | 11 | could | 55 |
| - | - | determine | 87 | potential | 10 | whether | 52 |
| - | - | putative | 80 | probable | 10 | perhaps | 51 |
| - | - | hypothesis | 77 | suggest | 10 | will | 39 |
| - | - | think | 66 | allege | 8 | appear | 32 |
| - | - | would | 52 | accuse | 7 | until | 15 |

Table 10: Most frequent speculation cues according to different domains/datasets. The statistics of SFU are from paper (Konstantinova et al., 2012) and other statistics are from paper (Szarvas et al., 2012).

| Top 1-10 | | Top 11-20 | | Top 21-30 | |
|---|---|---|---|---|---|
| Cue | Freq. | Cue | Freq. | Cue | Freq. |
| may | 1665 | appear | 260 | probable | 122 |
| if | 1294 | might | 256 | probably | 121 |
| or | 1017 | consider | 250 | study | 101 |
| suggest | 928 | whether | 247 | allege | 89 |
| can | 882 | seem | 228 | determine | 87 |
| would | 832 | investigate | 221 | putative | 80 |
| could | 495 | likely | 221 | hypothesis | 77 |
| indicate | 404 | should | 213 | expect | 75 |
| think | 362 | believe | 198 | will | 60 |
| possible | 304 | examine | 183 | regard | 58 |

Table 11: The 30 most frequent speculation cues by summing the frequency in Table 10.