# OpenReview forum: "From Speculation Detection to Trustworthy Relational Tuples in Information Extraction"
_EMNLP/2023/Conference — EMNLP 2023 Findings_

### Official Review · Reviewer_6jWm · 2023-08-03

**Typos Grammar Style And Presentation Improvements:** 1. There are some formatting issues a…
**Soundness:** 4

**Excitement:**

3: Ambivalent: It has merits (e.g., it reports state-of-the-art results, the idea is nice), but there are key weaknesses (e.g., it describes incremental work), and it can significantly benefit from another round of revision. However, I won't object to accepting it if my co-reviewers champion it.

**Paper Topic And Main Contributions:**

This paper focuses on speculation detection tasks but at a tuple granularity that has not been addressed before, rather than the common sentence granularity. This paper organizes the corresponding dataset for the new task and verifies the effect of the proposed model to a certain extent through experiments on it.

**Reasons To Accept:**

1. This paper focuses on the tuple granularity speculation detection task from a new perspective, which can be better applied to downstream tasks.
2. This paper organizes the corresponding dataset for the new task which can help further research in related fields.
3. Different dimensions deeply analyze the value of the newly proposed task and the effectiveness of the newly proposed method of the newly constructed dataset.


**Reasons To Reject:**

1. The experiment is not sufficient. Although it is a new task and there is no SOTA model, it should also be compared with the commonly used SOTA model for similar tasks.

**Reproducibility:**

5: Could easily reproduce the results.

**Reviewer Confidence:**

3: Pretty sure, but there's a chance I missed something. Although I have a good feel for this area in general, I did not carefully check the paper's details, e.g., the math, experimental design, or novelty.

---

### Official Review · Reviewer_Qo8K · 2023-08-03

**Soundness:** 3

**Excitement:**

4: Strong: This paper deepens the understanding of some phenomenon or lowers the barriers to an existing research direction.

**Paper Topic And Main Contributions:**

This paper introduces a novel task of tuple-level speculation detection. The objective is to determine whether a tuple extracted by an OIE system is speculative. This task offers a more precise refinement compared to previous sentence-level speculation detection tasks.

To obtain the speculation label of tuples, the authors relied on a naive conversion of the modal verb annotation from the LSOIE dataset. Specifically, if the predicate label in LSOIE includes any of the six modal verbs (might, can, will, would, should and had), the tuple is considered speculative.

Furthermore, the authors present a simple yet effective GNN baseline for tuple-level speculation detection. This baseline surpasses other naive baselines by a significant margin, providing a promising starting point for future research endeavours.

**Questions For The Authors:**

In lines 221-224, the author mentioned that they manually examined speculation labels.  What are the results of these examinations?

**Reasons To Accept:**

This paper puts forth an interesting OIE subtask that attempts to classify whether an extracted tuple is speculative. This often overlooked aspect of IE carries significant real-world implications. The method is quite simple, and it appears that the classification technique can be readily applied to other IE tasks, such as temporal IE. The proposed GNN baseline also offers a solid starting point for future research.

**Reasons To Reject:**

The main issue with this work is that the authors directly derived speculation labels from modality labels. While modality and speculation are closely related, there are several distinctions. For instance, the authors consider the modal verb "would" speculative; however, it also indicates a habitual event. For example, in the sentence "when I was an undergrad, I would fall asleep with music," the tuple _(I, would fall, asleep, with music)_ cannot be seen as speculative. Although the authors' direct conversion of modal verbs is an advantageous first step towards tuple-level speculation detection, it is necessary to label speculation information explicitly.

The authors acknowledged this issue [214-224; 614-627]. They argue since LOSIE does not primarily focus on explicitly labeling speculation information, it does not introduce bias towards any specific type of speculation [218-221]. However, I disagree with this particular claim. When the target of annotation differs, the first step should involve validating and aligning the two methodologies before discussing bias. The authors also mentioned they manually examined a large number of speculative labels [221-224], yet they failed to report any results from these examinations. Providing statistical evidence to support the claim that "the labels are reliable enough" would indeed be beneficial.

Minor points:
The citation Ellis (2022) is merely a blog post: https://www.grammarly.com/blog/auxiliary-verbs/.  It would be better to obtain definition of modal from peer-reviewed sources.


**Reproducibility:**

4: Could mostly reproduce the results, but there may be some variation because of sample variance or minor variations in their interpretation of the protocol or method.

**Reviewer Confidence:**

5: Positive that my evaluation is correct. I read the paper very carefully and I am very familiar with related work.

---

> ### Author Rebuttal · Authors · 2023-08-28
>
> 1. Regarding the potential bias of annotation of speculation labels [214-224; 614-627], we agree that validating and aligning two methodologies are important. However, it is costly to hire annotators to complete it. So we manually examnined a large number of labels (about 1,000 cases). Our examination focuses on difficult labels that are not present in the sentence. The checking is to ensure the speculation labels in the dataset do carry speculative meaning. Thanks for pointing it out, we will report more details in appendix in the revised version to report the relaiblility of speculative labels.
> 2. Thanks for the pointing out the citation of definition of model words, we will cite peer-reviewed sources instead of the blog post in our revised version.

---

### Official Review · Reviewer_hXdC · 2023-08-17

**Soundness:** 2

**Excitement:**

3: Ambivalent: It has merits (e.g., it reports state-of-the-art results, the idea is nice), but there are key weaknesses (e.g., it describes incremental work), and it can significantly benefit from another round of revision. However, I won't object to accepting it if my co-reviewers champion it.

**Paper Topic And Main Contributions:**

This paper proposes a new research task of detecting speculative information in Open Information Extraction (OIE) tuples. It highlights the impact of speculative information uncertainty on relational tuples, introducing a baseline model (SpecTup) based on semantic and syntactic modeling for tuple-level speculative detection. We analyze the existence and difficulty of tuple-level speculative labels, and verify the effectiveness of the proposed model on this task through experiments.

**Reasons To Accept:**

+ Novelty: This paper presents a new research task that bridges the gap between speculative detection and information extraction, emphasizing the impact of speculative information on the trustworthiness of relational tuples.

+ Effect: A comprehensive experimental analysis demonstrates the effectiveness of the proposed method.

**Reasons To Reject:**

- The large language model has achieved amazing capabilities in the NLP community. Whether it can also be solved well in this task, the author did not mention the application and effect of LLM. It is recommended to take this factor into consideration.

- This method combines semantic and grammatical information through different models, which is not uncommon, and the innovation of this method seems to be relatively small.

**Reproducibility:**

3: Could reproduce the results with some difficulty. The settings of parameters are underspecified or subjectively determined; the training/evaluation data are not widely available.

**Reviewer Confidence:**

3: Pretty sure, but there's a chance I missed something. Although I have a good feel for this area in general, I did not carefully check the paper's details, e.g., the math, experimental design, or novelty.

---

> ### Author Rebuttal · Authors · 2023-08-28
>
> 1. We agree that Large Language Model has achieved promising results on NLP tasks. The primary focus of our work is to propose a new research topic, rather than proposing a strong and sophiscated baseline to solve it. We will include some results of using LLM (e.g. Llama 2) in the appendix.
> 2. The fusion of syntactic information (dependency parsing tree) and semantic information (PLM: BERT) has been explored in some NLP models. The novelty of our fusion methodoloy is to design a novel relation-aware GCN that can adaptively aggregate neighbours based on different dependency relations. Moreover, existing methods focus on the syntactic structure of entire sentences, whereas our method focuses on the local syntactic information, i.e., speculation cue words.

---

### Official Review · Reviewer_g5sr · 2023-08-18

**Soundness:** 3

**Excitement:**

3: Ambivalent: It has merits (e.g., it reports state-of-the-art results, the idea is nice), but there are key weaknesses (e.g., it describes incremental work), and it can significantly benefit from another round of revision. However, I won't object to accepting it if my co-reviewers champion it.

**Paper Topic And Main Contributions:**

This paper presents an interesting research task: detecting the trustworthiness of tuples obtained from information extraction. The authors analyse the LSOIE dataset and find that about 10.9% of the labeled tuples contain speculative information. They model tuple-level speculation detection as a classification task and propose a detection model, SpecTup, that utilizes contextual semantic information and dependency structure information to classify speculative tuples. Comparison with keyword matching methods and ablation experiments demonstrate that the proposed method has better performance.


**Questions For The Authors:**

1. Why not use the superior RoBERTa model, which has a parameter count similar to BERT?

2. Is BERT fully fine-tuned in your approach, or is it just used as a feature extractor? How does the performance of directly fine-tuning BERT compare?

3. The authors mentioned they manually checked a large number of labels (lines 221-224). Are there more specific details available regarding this aspect?

**Reasons To Accept:**

1. This paper proposes an interesting research problem.

2. The authors design a strong baseline model and demonstrat that there is still a huge room for improvement in this task.

**Reasons To Reject:**

1. The proposed baseline model is complex and not easy to follow.

2. Limited dataset.

3. Limited comparison models, especially lacking comparison with the latest in-context learning methods.

**Reproducibility:**

3: Could reproduce the results with some difficulty. The settings of parameters are underspecified or subjectively determined; the training/evaluation data are not widely available.

**Reviewer Confidence:**

3: Pretty sure, but there's a chance I missed something. Although I have a good feel for this area in general, I did not carefully check the paper's details, e.g., the math, experimental design, or novelty.

---

> ### Author Rebuttal · Authors · 2023-08-28
>
> 1. The proposed baseline consists of three layers: (1) BERT, (2) GCN encoder, and (3) linear layer for binary/multi-class classification. Section 4 may be complicated without any illustration. We will include a diagram of the overall model architecture with the three layers in the paper.
> 2. The BERT module used in our baseline is fine-tuned during the training process. Rather explore many different PLMs such as RoBERTa, ELECTRA, T5, etc, we believe using BERT is sufficient for the exploration of the new tuple-level speculation. We did experiments on RoBERTa for the task, and found the experimental results are similar to BERT. So we only reported the results of using BERT, as tuple-level speculation is our key focus. We will include the results of RoBERTa in the appendix.
> 3. Regarding the manual checking of labels described in lines 221-224, our examination focuses on difficult labels that are not present in the sentence. The checking is to ensure the speculation labels in the dataset do carry speculative meaning. Thanks for pointing it out, we will report more details in the appendix in the revised version.

---

### Meta-Review · Area_Chair_PzWR · 2023-09-22

**Recommendation:** 2

**Metareview:**

All the reviewers have agreed that this paper has proposed a very interesting new task to detect the trustworthiness of extracted tuples.

However, many concerns have also been raised regarding to the experiments. And it seems that through in-depth rebuttal, these concerns remain to a large degree.

Generally, this paper anchors a new direction that worths further research.

---

### Decision · Program_Chairs · 2023-10-07

**Decision:**

Accept-Findings

**Comment:**

All the reviewers have agreed that this paper has proposed a very interesting new task to detect the trustworthiness of extracted tuples.

However, many concerns have also been raised regarding to the experiments. And it seems that through in-depth rebuttal, these concerns remain to a large degree.

Generally, this paper anchors a new direction that worths further research.